# Artificial light during the polar night disrupts Arctic fish and zooplankton behaviour down to 200 m depth

Jørgen Berge [1,2,3,10 ✉], Maxime Geoffroy[1,4,10], Malin Daase[1,10], Finlo Cottier[1,5,10], Pierre Priou [1,4], Jonathan H. Cohen[6], Geir Johnsen[2,3], David McKee [7], Ina Kostakis [7], Paul E. Renaud[2,8], Daniel Vogedes[1], Philip Anderson [5], Kim S. Last[5] & Stephane Gauthier[9]

For organisms that remain active in one of the last undisturbed and pristine dark environments on the planet—the Arctic Polar Night—the moon, stars and aurora borealis may provide important cues to guide distribution and behaviours, including predator-prey interactions. With a changing climate and increased human activities in the Arctic, such natural light sources will in many places be masked by the much stronger illumination from artificial light. Here we show that normal working-light from a ship may disrupt fish and zooplankton behaviour down to at least 200 m depth across an area of >0.125 $km^2$ around the ship. Both the quantitative and qualitative nature of the disturbance differed between the examined regions. We conclude that biological surveys in the dark from illuminated ships may introduce biases on biological sampling, bioacoustic surveys, and possibly stock assessments of commercial and non-commercial species.

[1] Department Arctic and Marine Biology, Faculty for Bioscience, Fisheries and Economy, UiT The Arctic University of Norway, Tromsø, Norway. [2] Department of Arctic Biology, University Centre in Svalbard, Longyearbyen, Norway. [3] Center of Autonomous Marine Operations and Systems, Department of Biology, Norwegian University of Technology and Science, Trondheim, Norway. [4] Centre for Fisheries Ecosystems Research, Fisheries and Marine Institute of Memorial University of Newfoundland, St. John's, NL, Canada. [5] Scottish Association for Marine Science, Oban, UK. [6] School of Marine Science and Policy, University of Delaware, Lewes, DE, USA. [7] Physics Department, University of Strathclyde, Glasgow, UK. [8] Akvaplan-niva, Fram Center for Climate and the Environment, N-9296 Tromsø, Norway. [9] Fisheries and Oceans Canada, Institute of Ocean Sciences, Sidney, BC V8L 4B2, Canada. [10] These authors contributed equally: Jørgen Berge, Maxime Geoffroy, Malin Daase, Finlo Cottier. ✉email: jorgen.berge@uit.no

Artificial illumination at night is increasing annually by 6% on average, thus becoming one of the fastest-spreading environmental challenges of the Anthropocene[1]. An estimated 23% of all land masses between 75°N and 60°S and 22% of all coastal regions are now believed to be exposed to scattered artificial light that is reflected back from a cloud rich atmosphere[2,3], while artificial lights from cities, coastlines, roads and marine infrastructures are visible from outer space. The study of environmental impacts of artificial light has been a rapidly growing field in recent years[2,4,5].

We now know that light pollution, hereafter artificial light, affects both organismal behaviour and ecosystem processes across a wide range of taxonomic groups[2] and ecosystems[6,7]. Aquatic examples of artificial light impact include effects on primary production and community structure[4], especially when exposed to white LED (light emitting diodes) light, with affected areas ranging from coastal regions to offshore platforms (reviewed by Davies et al.[5]). Other well-known examples of how artificial light affects organisms include the disorientation experienced by species that use natural light cues to orient or navigate, most notably sea turtles and birds[8,9]. Even in sparsely populated areas of the Arctic, artificial light is starting to become noticeable, particularly in coastal areas[10]. Moreover, in a future warmer Arctic with a dramatically reduced ice cover, human activities and footprints are likely to increase. While changing temperature, pH, ice cover, and $CO_2$ levels are all factors that have affected biological communities throughout their evolutionary history, artificial light is unprecedented. Thus, no species has had the need or opportunity to evolve in relation to artificial light. On the contrary, artificial light competes with the harmonic movements of the earth, moon and sun that provide reliable photoperiodic cues to which behaviour and physiology are highly attuned[11,12].

Recent advances in the study of Arctic marine ecosystems during the Polar Night have caused a radical shift in how we perceive their seasonality and function[13,14]. Instead of an ecosystem that enters a resting state during the winter darkness, we now recognise a highly active ecosystem characterised by continuous activity and biological interactions across all trophic levels and taxonomic groups[13,14]. Within what is referred to as Nautical Polar Night, there is (to the human eye) virtually no difference in illumination between night and day[13]. Nevertheless, many marine organisms stay active and are able to adjust their behaviour to the diel cycle of background solar illumination[13-15]. Examples of processes known to be regulated by small changes in natural ambient light include diel vertical migration both under sea ice[16] and in the upper 30 m of the water column during nautical Polar Night[15,17,18], lunar impacts on behaviour[19], and trophic interactions influenced by bioluminescence in shallow waters[18]. Mesozooplankton have also been shown to respond to artificial light from a ship down to 100 m depth[15]. Euphausiids (krill) on the Nova Scotia continental shelf avoid artificial light from ships at night[20], and artificial light is also known to both attract and repel fish[21,22], depending on the species. The dark Polar Night, however, remains a major knowledge gap regarding both natural behaviour and, not the least, potential effects of artificial lights at sea.

In a system where organisms remain active and are adapted to detect and respond to extremely low levels of natural light during the Polar Night, we postulate that their susceptibility towards artificial light is likely to be high. With a continued warming and reduction of Arctic sea ice, human presence and activity in the region are predicted to increase substantially[23]. Inevitably, so will artificial light, for instance from ship traffic, fisheries, and oil and gas activities. If our assertion concerning enhanced susceptibility of Arctic organisms towards artificial light during the Polar Night is correct, this represents a key challenge for future sustainable use, human presence, and sustainable harvesting of marine resources in the Arctic. Here, we investigate how artificial light from a ship affects the vertical distribution of macrozooplankton and pelagic fish communities around the ship, and assess the potential for bias in biological surveys carried out from ships artificially illuminating an otherwise dark environment. To understand how different pelagic communities respond to artificial light, we quantify the differences in the acoustic backscatter measured within the water below a ship that is fully illuminated ('lights on') and in complete darkness ('lights off') under different environmental conditions and for different species assemblages.

We found that fish and macrozooplankton communities exhibit an almost instantaneous response that reached 200 m depth (limited only by water depth) when exposed to artificial light. The quantitative and qualitative nature of the disturbance varied with physical and biological characteristics, suggesting that extreme caution must be taken when conducting scientific surveys or stock assessments with artificial light in the Arctic Polar Night.

## Results

**Differences in sampling locations**. We conducted three field experiments using a combination of acoustics, trawls and zooplankton nets from the same ship during the course of 1 week in January 2018 at three contrasting stations (Fig. 1, left panel). The northern two stations (A and B, Fig. 1) experienced astronomical[24] and nautical twilight[24], respectively, while the southern station (C, Fig. 1) experienced higher levels of ambient light around solar noon, even though the sun was still below the horizon at all times. At all three stations, the normal working lights from the ship providing an irradiance ($E_{PAR}$) of 2.2 μmol photons $m^{-2} s^{-1}$ at the sea-surface, directly beneath the lights, affected the fish and microzooplankton communities. The sites (A–C) varied not only in latitude (and hence also ambient light climate), but also in their hydrographic characteristics (Fig. 1a–c) and species composition (Table 1).

**Light-induced impacts in the water column**. Changes in the acoustic backscatter of pelagic organisms in response to artificial light from the ship were different at each of the three stations (Fig. 2). These responses were tested by turning on the ship's lights after a period of total darkness. At stations A and C, the total acoustic backscatter was reduced by 47–54% and 4–19%, respectively, while at station B, the response was the opposite, with a 43–55% increase in acoustic backscatter (Table 2). Krill, polar cod and Atlantic cod dominated the assemblage at station A and northern shrimp and Atlantic herring dominated at station B (Table 1). For a given assemblage, the relationship between acoustic backscatter (i.e., Nautical Area Scattering Coefficient[25] (NASC) in $m^2 nmi^{-2}$) and biomass is linear and such increases or decreases in acoustic backscatter result in equivalent biases in biomass estimates[25]. The change in vertical distribution of fish and zooplankton was relatively limited (i.e., variation in centre of mass <10 m) at all three sites (Table 2), suggesting that the instantaneous decrease or increase in backscatter is first related to a change in orientation rather than a rapid descent or ascent. These differences in response suggest that there is no unequivocal way of characterising the effect of artificial light, but that knowledge about both physical and biological factors are key to predicting the effect of artificial light from a ship on its surroundings.

**Ambient and artificial light**. Average absorption (a) and light backscatter ($b_b$), both at 498 nm, over the top 20 m of the water column varied from 3.48E-02 to 6.61E-02 $m^{-1}$ and 1.99E-03 to 1.76E-02 $m^{-1}$, respectively (Table 3). Corresponding estimates of the diffuse attenuation coefficient $K_d$(489 nm) varied from

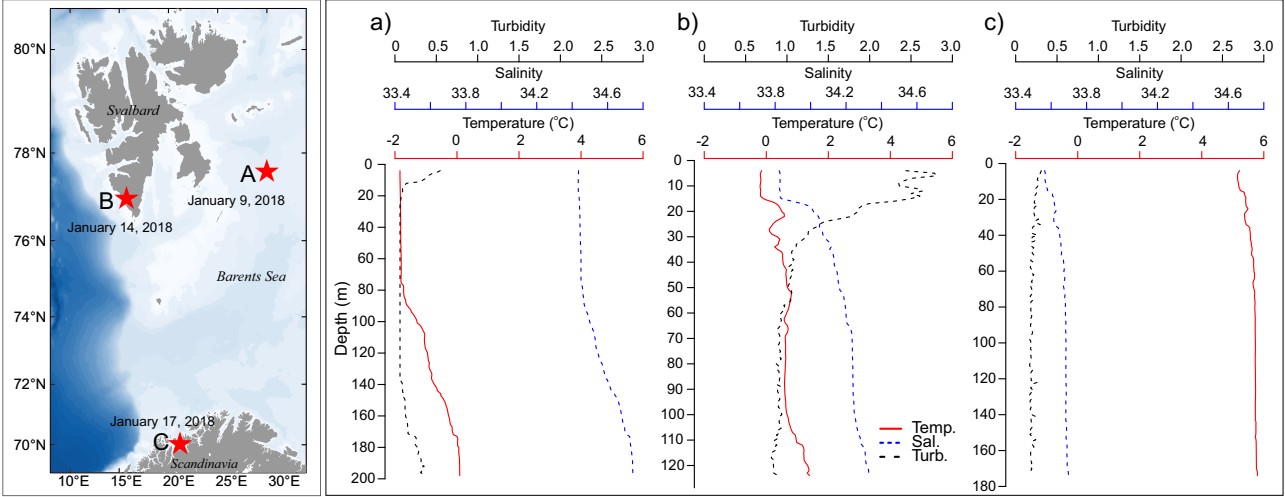

**Fig. 1 Study area and hydrography characteristics of the three study sites.** Map of the study area indicating the position and date of the three field experiments (A, B and C). **a–c** show hydrographic characteristics: turbidity (formazin turbidity units, FTU), salinity and temperature, all as a function of depth, at the corresponding locations.

**Table 1 The ten most abundant species (in terms of biomass) in the bottom trawl (BT) and midwater trawl (MWT) at stations A and B.**

| | Station A | | | Station B | | |
|---|---|---|---|---|---|---|
| | **Species name** | **Count (n)** | **Weight (g)** | **Species name** | **Count (n)** | **Weight (g)** |
| MWT | *Thysanoessa* spp. | 26060 | 2343 | *Clupea harengus* | 6421 | 23,912 |
| | *Mallotus villosus* | 685 | 2142 | *Pandalus borealis* | 1060 | 3914 |
| | *Cyanea capillata* | 2 | 181 | *Boreogadus saida* | 1142 | 2935 |
| | *Themisto libellula* | 598 | 141 | *Gadus morhua* | 29 | 723 |
| | Hydrozoae | 16 | 136 | *Leptoclinus maculatus* | 374 | 480 |
| | *Meganictyphanes norvegica* | 337 | 117 | *Liparis gibbus* | 53 | 373 |
| | *Boreogadus saida* | 10 | 22 | *Melanogrammus aeglefinus* | 8 | 202 |
| | *Clione limacina* | 89 | 17 | *Hippoglossoides platessoides* | 4 | 132 |
| | *Sebastes* spp. | 9 | 16 | *Mallotus villosus* | 15 | 60 |
| | *Leptoclinus maculatus* | 6 | 11 | *Sabinea septemcarinata* | 9 | 50 |
| BT | *Gadus morhua* | 218 | 22376 | *Pandalus borealis* | 5714 | 29,556 |
| | *Boreogadus saida* | 578 | 8367 | *Hippoglossoides platessoides* | 409 | 14,600 |
| | *Mallotus villosus* | 262 | 2506 | *Amblyraja* sp. | 3 | 2066 |
| | *Hippoglossoides platessoides* | 59 | 2364 | *Boreogadus saida* | 136 | 1569 |
| | *Reinhardtius hippoglossoides* | 4 | 1706 | *Reinhardtius hippoglossoides* | 4 | 1180 |
| | *Pandalus borealis* | 265 | 1490 | *Gadus morhua* | 182 | 426 |
| | *Sebastes mentella* | 52 | 1367 | *Mallotus villosus* | 59 | 407 |
| | *Lycodes reticulatus* | 32 | 1062 | *Leptoclinus maculatus* | 81 | 388 |
| | *Sabinea septemcarinata* | 131 | 476 | *Melanogrammus aeglefinus* | 9 | 255 |
| | *Leptoclinus maculatus* | 91 | 368 | *Clupea harengus* | 36 | 214 |

Trawling was not permitted at station C. Depth of trawls: MWT 128 m and BT 200 m at station A, MWT 50 m and BT 140 m at station B.

4.01E-02 to 1.17E-01 m$^{-1}$. The profound effect of these seemingly small differences in diffuse attenuation on the underwater light field are illustrated by calculation of the fraction of irradiance at 489 nm that would reach the maximum depth of our study 200 m, with P200 varying from 6.77E-09 to 3.29E-02 % (Table 3). Station A presented the clearest water resulting in as much as 0.03% of the surface irradiance at 489 nm reaching as far as 200 m depth. Integrating across the visible spectrum (400–700 nm) gives an order of magnitude lower value for visible light, reflecting higher attenuation at other spectral bands. We note that small increases in absorption and light backscatter at the other two stations (maximally doubling) result in orders of magnitude decreases in irradiance levels for equivalent depths. Despite being almost 100 m deeper, light levels at the seabed would be more than an order of magnitude higher at Station A than at Station B.

Ambient light levels (with ship's lights switched off) at stations A and B were sufficiently dark as to be below the measurement threshold of the available Trios hyperspectral spectroradiometers (0.0004 mW m$^{-2}$ nm$^{-1}$). However, at station C, ambient light levels at noon were 5.28 mW m$^{-2}$ nm$^{-1}$ at 489 nm, equivalent to $E_{PAR}$ 12.3 µmol photons m$^{-2}$ s$^{-1}$, or five orders of magnitude higher than at the two stations further north. Importantly, all experiments reported here were conducted duing periods of darkness, when ambient light levels were below our measurement threshold and observed light signals are attributable to ship's lights. When lights were switched on, the light field emitted by the ship produced a highly variable three-dimensional distribution around the ship, with maximum values found close to the illuminated ship's CTD (Conductivity, Temperature, Depth profiler) hatch.

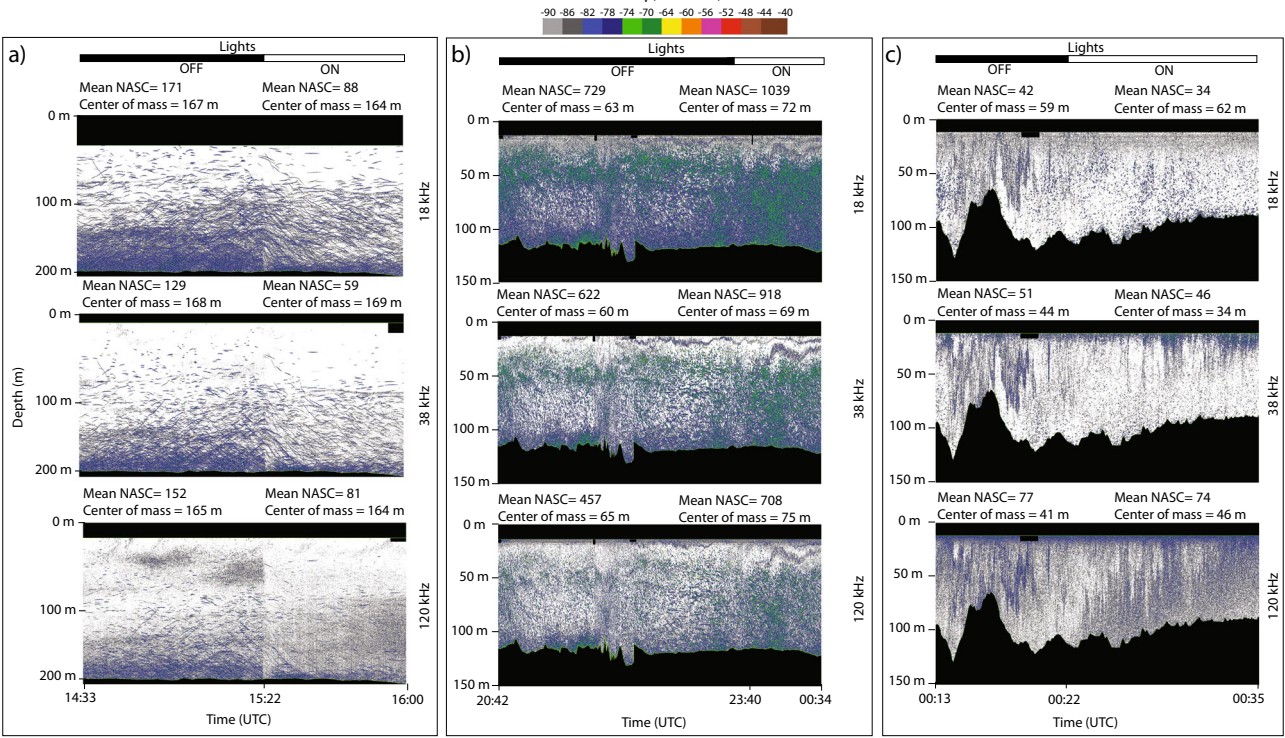

**Fig. 2 EK60 echograms at 18, 38 and 120 kHz when lights are OFF and ON.** EK60 echograms of volume backscattering strength ($S_v$) at 18 and 38 kHz dominated by the signal of fish and macrozooplankton and 120 kHz dominated by meso- and macrozooplankton at stations A (**a**), B (**b**) and C (**c**). The nautical area backscattering coefficient (NASC in $m^2\,nmi^{-2}$) and centre of mass with lights ON and OFF are indicated for each echogram, see also Table 2. Light levels emitted from the ship (lights ON) was 0.45 mW m$^{-2}$ nm$^{-1}$ at 489 nm just below the surface, equivalent to $E_{PAR} = 2.24$ µmol photons m$^{-2}$ s$^{-1}$.

**Table 2 The nautical area backscattering coefficient (NASC in $m^2\,nmi^{-2}$) and centre of mass (CoM) at each of the three stations.**

| | | Station A | | | Station B | | | Station C | | |
|---|---|---|---|---|---|---|---|---|---|---|
| | | Lights off | Lights on | Δ | Lights off | Lights on | Δ | Lights off | Lights on | Δ |
| 18 kHz | NASC | 171 | 88 | −49% | 729 | 1039 | +43% | 42 | 34 | −19% |
| | CoM | 167 | 164 | −3 m | 63 | 72 | +9 m | 59 | 62 | +3 m |
| 38 kHz | NASC | 129 | 59 | −54% | 622 | 918 | +48% | 51 | 46 | −10% |
| | CoM | 168 | 169 | +1 m | 60 | 69 | +9 m | 44 | 34 | −9 m |
| 120 kHz | NASC | 152 | 81 | −47% | 457 | 708 | +55% | 77 | 74 | −4% |
| | CoM | 165 | 164 | −1 m | 65 | 75 | +10 m | 41 | 46 | +5 m |

Data are from the hull-borne echosounder with three frequencies; 18, 38 and 120 kHz. Values are calculated as a mean value for the entire water column during the entire period the lights were on/off.

**Table 3 Absorption, $a$, light backscatter, $b_b$, and diffuse attenuation coefficients, $K_d$, at 489 nm for stations A, B and C.**

| Station | $a489$ (m$^{-1}$) | $b_b489$ (m$^{-1}$) | $K_d489$ (m$^{-1}$) | $E_{PAR}$bottom (%) | $E_{PAR}$200 (%) |
|---|---|---|---|---|---|
| A | 3.48E-02 | 1.99E-03 | 4.01E-02 | 3.14E-03 | 3.14E-03 |
| B | 8.98E-02 | 1.76E-02 | 1.17E-01 | 1.45E-05 | 1.61E-09 |
| C | 6.61E-02 | 5.53E-03 | 7.81E-02 | 7.32E-05 | 2.70E-06 |

The fraction of 489 nm irradiance penetrating from surface to depth was calculated using Eq. 1, while the percentage penetration of photosynthetically available radiation to bottom ($E_{PAR}$bottom) and 200 m ($E_{PAR}$200) was obtained by integrating irradiance data over 400–700 nm, providing $E_{PAR}$.

Downward planar irradiance at 489 nm (the wavelength of greatest penetration of light with depth in this region) immediately beneath the sea-surface at this position was found to be 0.45 mW m$^{-2}$ nm$^{-1}$, corresponding to an equivalent $E_{PAR}$ (photon scale) value of 2.2 µmol photons m$^{-2}$ s$^{-1}$. Note that the apparent difference in conversion factor between energy (mW) and quanta (µmol photons) between this and the conversion of ambient light above is due to differences in spectral distributions measured in situ.

**Spatial impact of artificial light**. We measured the horizontal distance from the ship at which artificial light had an impact on fish and zooplankton backscatter using a portable echosounder

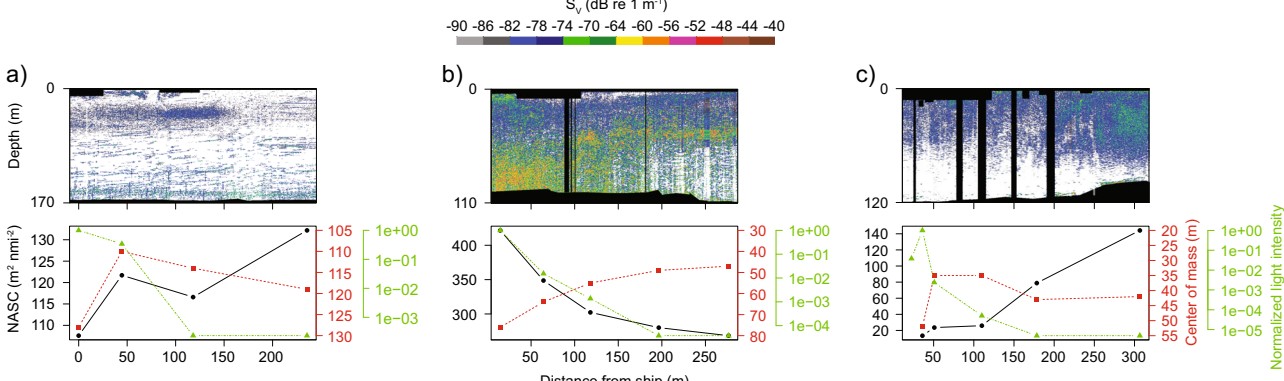

**Fig. 3 Changes in acoustic backscatter and light intensity with distance from an illuminated ship.** Echograms of volume backscattering strength ($S_v$) at 125 kHz and corresponding variations in NASC (solid black line), centre of mass (dashed red line) and normalise light intensity (dashed green line) measured from a small open boat fitted with a portable echosounder as it moved away from the illuminated ship at stations A (**a**), B (**b**) and C (**c**).

deployed on the side of a small boat. At Station A, the backscatter (calculated as NASC) diminished by 12% and the centre of mass descended by 18 m within the first 50 m from the ship. No measurable effects were noted beyond that distance although the light footprint reached 120 m away from the ship (Fig. 3a). At Station B, both the light footprint and the measurable changes in backscatter and centre of mass reached 200 m. The NASC increased by 57% and centre of mass descended by 27 m (Fig. 3b). At Station C, the light footprint reached 180 m and the back-scatter within that range diminished by 83%. However, the centre of mass only diminished within 50 m from the ship (descent of 17 m) (Fig. 3c). The different responses to artificial light measured from one station to the other correspond to the observations from the hull-mounted EK60, with a decrease in backscatter at stations A and C, dominated by gadoids and krill, and an increase at Station B where northern shrimp and herring were abundant.

## Discussion

At the first site (A), a decrease in acoustic backscatter was detected across the entire water column down to 200 m depth as the lights on the ship were turned on (Fig. 2). As this occurred through the entire water column, only limited by the maximum water depth at this particular station, the results suggest that the potential impact zone of artificial light from the ship extends well into the mesopelagic layer (i.e., beyond 200 m depth). At the other two stations, a comparably deep response was not detected. At Station B, higher turbidity limited light penetration at depth (Fig. 1 and Table 3). At both stations B and C, light levels were two orders of magnitude lower at the bottom compared to Station A (Table 3). The theoretical retention of light at 200 m depth at Station B was six orders of magnitude lower than at Station A (Table 3). At Station C, the ambient light at noon is characterised as civil twilight[24] with the sun remaining below the horizon throughout the diel cycle. At stations A and B, ambient light at noon are defined as astronomical and nautical twilight, respectively, with maximum ambient atmospheric light levels up to five orders of magnitude lower than at Station C[24]. In accordance with our assertion that there is an enhanced susceptibility towards artificial light at higher latitudes, the relative change in mean NASC (Table 2) was lowest at the southernmost station, where organisms may be less sensitive to changes in background illumination. Yet, the relative change in mean NASC at station C was −19% (Table 2), indicating that artificial light also affected organisms at the lower latitude station.

The lower acoustic backscatter at stations A and B and higher backscatter at Station C measured from the small boat

compared to the research ship likely results from temporal variations, as experiments were decoupled in time to avoid acoustic interferences. Regardless, in both cases (measurements from the research ship and small boat) we obtained NASC values within the same order of magnitude (Figs. 2 and 3). Temporal variations and patchiness of organisms could also partly explain the slightly lower centre of mass under the ship with lights turned off (Fig. 2) compared to values outside the light footprint measured with the small boat (Fig. 3), for instance 65 m vs. 47 m at Station B. However, this discrepancy might also be related to fish and zooplankton avoiding noise from the research ship, in addition to light[26]. If so, even with lights off the vertical distribution of fish and zooplankton measured with the hull-mounted echosounder would have been lower than the undisturbed state. However, with the small boat (with engine and lights turned off), we would have observed the 'natural' state of the ecosystem. Based on the acoustic mea-surement showing the backscatter being affected up to 200 m away from the ship, we suggest that artificial light from a ship has the potential to affect acoustically detectable biomass within an area of >125,000 m² around that ship.

Artificial light caused an almost immediate response (within 5 s) in the pelagic community throughout the entire water column down to at least 200 m depth and up to 200 m away from the ship. Our results show that the effects of artificial light on fish and macrozooplankton, potentially extending down to the mesope-lagic layer, represent a key challenge for future sustainable use and development of marine resources in the Arctic. First, increased artificial light will have a direct impact on organisms, their vertical positioning in the water column and their swim-ming behaviour. Second, it will affect our capacity to understand these processes as research surveys and acoustic stock assess-ments carried out using illuminated ships in the dark Polar Night are likely biased (Fig. 4).

Our results also raise questions regarding the potential effects of artificial light on biomass derived from acoustic backscatter measurements carried out in the dark outside the Arctic region. Lights introduced by different types of platforms could potentially impact surveys that are conducted at (or near) nighttime, parti-cularly those targeting small pelagic species that are distributed in relatively shallow waters. For example, spring surveys of capelin stocks in eastern Canada, occur to 25% during nighttime[27,28], as do acoustic surveys on Atlantic herring in various regions[29–31]. Potential differences in acoustic backscatter response between day and night have been discussed for Atlantic herring (see ref. [32]). Our study revealed a reduction in backscatter of 47–54% for capelin and an increase of 4–19% for herring when exposed to

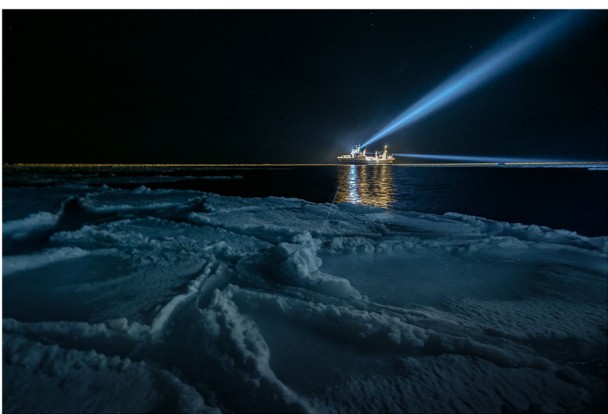

**Fig. 4 A research ship with normal working lights turned on deep inside the darkest Polar Night.** This paper examine the effects of artificial lights from this research vessel on its immediate environment. Our results raise questions regarding how relevant any biological data collected from this ship will be regarding ecosystem processes, stock assessments or organismal behaviour.

artificial light during dark conditions. However, the degree to which these results are relevant for regions outside the Arctic remains unknown.

Strong diel effects have been documented for species that are distributed at much greater depths, such as blue whiting[33], another important commercial species that is also surveyed during both daytime and nighttime. Diel differences in acoustic estimates of several demersal species have also been investigated[34]. As observed here, these studies largely attribute changes in acoustic responses to changes in tilt angle distribution and consequent target strength during the ascent and descent of diel vertical migration, rather than avoidance of artificial light. Similar conclusions were drawn for mesopelagic fish in the Southern Ocean[35]. The impact of artificial light on pelagic fish surveys is complicated and confounded by other behavioral effects that have important survey considerations. As many species exhibit daytime schooling behaviour and perform notable diel vertical migration near the surface at night, many acoustic surveys are conducted only during daylight hours. However, schooling behaviour in pelagic species is highly variable, and for some species (e.g., jack mackerel sometimes forming dense schools at night), it has been shown to be functional and responsive to prey availability, not simply diel cycles[36]. However, even in cases where strong day-night differences in aggregating patterns are fairly consistent and only daytime acoustic data are considered, survey operating rules are often not clearly defined or enforced. This could lead to sampling of crepuscular periods, particularly for species that are known to exhibit diel migrations, leading to stock estimation biases. For example acoustic surveys of Antarctic krill often operate from 'local sunrise to local sunset'[37,38], while onset and effects of diel behaviour can extend well before or beyond these times. Organisms may be particularly responsive to surface artificial light during these periods.

We conclude that artificial light has the potential to affect the behaviour of marine fish and zooplankton during the Arctic polar night down to at least 200 m depth, but that the quantitative and qualitative nature of the response can vary. As a consequence, the precise nature of the light response can only be assessed and predicted based on a thorough understanding of the physical and biological characteristics of the impacted area. Based on the results presented herein, and the clear difference in how organisms reacted to artificial light from a ship, we suggest that light introduced by survey platforms is an additional factor that can

exacerbate (or introduce) differences in acoustic backscatter between daytime and nighttime and generate important biases. Fisheries, scientific explorations and pelagic stock assessments are expanding northward and into the Polar Night. With the potential effects of artificial light from ships extending into the mesopelagic layer, artificial light will require special consideration for the sustainable management of Arctic and sub-Arctic regions.

## Methods

For each of the three experiments we used the RV *Helmer Hanssen* as our main research platform. During the experiments, the ship was drifting with the engine on but without any power on the propellers. Also, during all three experiments, the same level of noise, light field/intensity and activity were ensured in order to dismiss these factors as responsible for any difference in response between each experiment.

**Species composition**. At Station A, krill (*Thysanoessa* spp.) and capelin (*Mallotus villosus*) dominated the pelagic community, whereas Atlantic cod (*Gadus morhua*) and polar cod (*Boreogadus saida*) were the two most dominant taxa in the bottom trawl (Table 1). The Target Strength of these species have been shown to decrease when their tilt angle increases[39], for instance to dive away from surface illumination. The zooplankton community was dominated by copepods (mainly *Calanus* spp.) accounting for 98% of the community (total zooplankton abundance 22.7 (+/−6.9) ind. m$^{-3}$). At Station B, northern shrimp (*Pandalus borealis*) and juvenile long rough dab plaice (*Hippoglossoides platessoides*) dominated in the bottom trawl while herring (*Clupea harengus*) and northern shrimp dominated the midwater trawl assemblage. Herring has been shown to react in a distinct way when exposed to artificial light, both as they pack in dense schools and by being attracted to artificial light[40]. In contrast to krill and fish, change in the tilt angle of northern shrimp, for instance to avoid light at the surface, alter their target strength in a way that increases their acoustic target strength[41]. Zooplankton was abundant at Station B (73.4 (+/−10.6) ind. m$^{-3}$) with the majority consisting of *Calanus* spp. (87%), krill (*Thysanoessa inermis*, 4%) and chaetognaths (*Parasagitta elegans* 5%). Trawling was not permitted at Station C, so at this location we do not have direct information regarding species composition of fish and larger macrozooplankton. Zooplankton net samples showed that the community consisted to 81% of copepods (*Calanus* spp., *Metridia longa*), as well as krill (5%) and chaetognaths (8%), but abundance was very low (1.6 (+/−0.6) ind. m$^{-3}$).

**Survey design**. Artificial light experiments were conducted in situ from the RV *Helmer Hanssen* offshore the East coast of Spitsbergen (77°33.5'N 29°59.9'E) on 9 January, 2018, in Hornsund (76°56.3'N 16°15.8'E) on 14 January, 2018, and North of Tromsø (70°06.1'N 19°16.71'E) on 17 January, 2018 (Fig. 1, left panel). Lights used were normal working lights representative for any ship operating in the dark. All lights from the *Helmer Hanssen* were turned off for 49 min (9 January), 178 min (14 January) or 9 min (17 January) before being turned on again. Change in the acoustic backscatter was recorded from the hull-mounted EK60 echosounder (18, 38, and 120 kHz). The ping rate was set to maximum and pulse length to 1024 µs. The echosounder was calibrated using the standard sphere method[42]. A Seabird 911 Plus CTD® fitted with a Seapoint Turbidity sensor recorded temperature, conductivity and turbidity during each experiment.

To measure the spatial impact of artificial light footprint from the ship, an Acoustic Zooplankton and Fish Profiler (AZFP 38, 125, 200, 455 kHz; ASL Environmental Science, Victoria, Canada) was deployed from a small boat (Polarcirkel™) stationary, but at varying distances from the *Helmer Hanssen*. For the AZFP, we only analysed the data at 125 kHz because higher frequencies have a limited range and the 38 kHz dataset was corrupted by the hull of the Polarcirkel due to wide side lobes. Vertical resolution varied from 37 cm on 9 January, and 2 cm on 14 and 17 January. The pulse duration was 1000 µs, ping rate 0.33 Hz (i.e., 1 ping 3 s$^{-1}$) and source level was 210 dB (re 1 µPa at 1 m). The AZFP was calibrated by the manufacturer (± 1dB) prior to deployment. The AZFP and EK60 echosounder on the *Helmer Hanssen* were operated at the same stations, but not at the same time to avoid interferences.

**Acoustic analyses**. Acoustic data were scrutinised and cleaned with Echoview® 8. We used Echoview's algorithms to remove background and impulse noise from EK60 and AZFP data, and attenuated noise signals from AZFP data[43,44]. The echograms at 18, 38, 120 kHz (EK60) were divided in 1 min long x 3 m deep echo integration cells[25]. The mean integrated Nautical Area Backscattering Coefficient (NASC in m$^2$ nmi$^{-2}$) from the hull-mounted echosounder was compared with lights on and off for each station. We calculated the centre of mass of the backscatter to document changes in vertical distribution[45]. For the AZFP, we stopped for 5 min near RV *Helmer Hanssen*, then at 50 m and afterwards every 100 m until 200 m (9 January), or 300 m (14 and 17 January). The centre of mass and total NASC at each stop were calculated at 125 kHz to assess the distance at which the lights from the ship impact the acoustic backscatter (i.e., proxies for the depth and biomass of scatterers).

**Fish and zooplankton sampling**. We deployed a Harstad pelagic trawl and a Campelen bottom trawl at stations A and B to groundtruth the acoustic signal. For safety reasons, the trawling was carried out with working lights turned on before the light experiments were initiated. All necessary licences and approvals were secured to carry out the trawling, which were always kept to an absolute minimum period. The Harstad trawl had an opening of $18.28 \times 18.28$ m and an effective height of 9–11 m and width of 10–12 m at three knots. The Campelen was trawled at three knots from 20 min and had an opening of 48–52 m, and a codend with an inner-liner mesh of 10 mm. All organisms were identified to species or genus onboard and we recorded the total number and weight of each species. Unfortunately, no trawl was deployed at Station C.

At Station A, we used a WP3 net (Hydrobios Kiel, 1 m$^2$ opening 1 mm mesh size) to take three zooplankton samples from 50 to 0 m. At stations B and C, zooplankton was sampled using a MIK net (Methot, Isaacs, Kidd Midwater Ring Net, 3.14 m$^2$ opening, 14 m long with main net bag of 1.5 mm mesh size, and the terminal 1.5 m part of 0.5 mm mesh size). Six vertical hauls were taken from 70 to 0 m at each station. All zooplankton sampling were conducted with working lights on the ship turned off. The samples were preserved in 4% hexamethylenetetramine-buffered seawater formaldehyde solution immediately after collection. In the laboratory, larger organisms (e.g., krill, chaetognaths, jellies) were identified and enumerated from subsamples using a plankton splitter (Station B) or from the entire sample (stations A and C). Copepods were counted from subsamples using a macropipette. Subsampling was continued until at least 300 individuals per sample were enumerated. Abundance estimates (individuals m$^{-3}$) are based on filtered water volume measured by a flowmeter attach to the centre of the MIK net opening, and for the WP3 samples by multiplying mouth opening area of the net by vertical hauling distance assuming 100% filtration efficiency.

**Light measurements**. Absorption and light backscatter profiles were recorded at 9 wavebands across the visible spectrum using WETLabs AC-9 (light beam attenuation metre) and BB9 optical sensors, respectively. Data from both instruments were corrected for light absorption and scattering artefacts following standard manufacturer's correction methods. The AC-9 was calibrated using freshly drawn Milli-Q ultrapure water on board the ship. Temperature and salinity corrections were applied using concurrent data from Seabird SBE19Plus CTD profiles. Irradiance from ship's lights at the sea-surface was measured using a hyperspectral Trios RAMSES planar irradiance sensor, giving $E_{PAR} = 2.24$ µmol photons m$^{-2}$ s$^{-1}$ just beneath the surface. Retention of light (relative values) were measured from the open boat using a set of specially designed sensors to measure light in situ during the Polar Night[24].

The penetration of light at a given wavelength into the water column was calculated using the Beer-Lambert Law

$$E(z,\lambda) = E(0^-,\lambda)\exp[-K_d(\lambda)z] \tag{1}$$

where the diffuse attenuation coefficient, $K_d(\lambda)$, was estimated from measurements of light absorption, $a(\lambda)$, and backscattering, $b_b(\lambda)$, using the relationship from[46]

$$K_d(\lambda) = \frac{g[a(\lambda) + b_b(\lambda)]}{\mu_d} \tag{2}$$

Here, the parameter $g = 1.0395$ and $\mu_d$ is the mean cosine for downwards irradiance and is estimated assuming light source at zenith ($\theta = 90°$) and the relationship[47]

$$\mu_d = 0.827 \cos\theta_{sw} + 0.144 \tag{3}$$

**Statistics and reproducibility**. Statistical analyses (i.e., mean NASC) were conducted using RStudio Version 1.1.442. Echoview dataflow and Matlab code used for acoustic and light analyses, respectively, are available upon request.

**Reporting summary**. Further information on research design is available in the Nature Research Reporting Summary linked to this article.

## Data availability

All acoustic and light measurement data are available on the Polar Data Catalogue (https://www.polardata.ca/) under access code CCIN 13104.

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

## Acknowledgements

The work was supported by three grants from the Norwegian Research Council: *Arctic ABC* (project no 244319, *Arctic ABC Development* (project no 245923), *Deep Impact* (project no 300333), and the Centre of Excellence *AMOS* (project no 223254). We would like to thank the crew onboard RV *Helmer Hanssen* for all their efforts and support during the field campaign. This is a contribution to the Arctos research network (arctos.uit.no), Arctic Science Partnership (www.asp-net.org), the Ocean Frontier Institute funded through the Canada First Research Excellence fund, and to the ArcticNet project ArcticFish.

## Author contributions

J.B., M.G., M.D. and F.C. contributed in the planning and performance of the field experiments, analyses of data and writing the manuscript. P.P., J.H.C., G.J., D.M., I.K., P. E.R., K.L. and S.G. contributed in performance of the experiments, analyses of the data and writing the manuscript. P.A. contributed in analyses of the data. D.V. contributed in the planning and performance of the field experiments.

## Competing interests

The authors declare no competing interests.
