## [Peer Review File · Communications Biology]

Reviewers' comments:

Reviewer #1 (Remarks to the Author):

This study addresses a highly relevant issue of marine ecosystem research and provides new data from the in this respect under-studied Arctic Ocean. I consider it a highly valuable and timely contribution to science with multiple implications, not the least on the on-going year-round ice drift study MOSAiC. The hydroacoustic methodology and the presented analyses are solid. However, the sampling of zooplankton communities is of limited value to support the hydroacoustic data, because two very different types of zooplankton nets were used. Furthermore, the fish community could not be sampled at station C.

My only concern with this study is that – given the differences in direction and magnitude of the postulated effect of light on pelagic animal assemblages – the sample size is relatively low, and limited to the shallow habitat of the Barents Sea / Svalbard shelf. It would be quite relevant to know the vertical dimension on this effect, and how pelagic deep-ocean communities would react to artificial illumination. Furthermore, a really strong effect was only observed at one station (Station A). To support the postulated general bias of sampling from illuminated ships, one would like to see this effect at more than one station to exclude that an unknown and unintended factor rather than light caused the observed dispersion of backscatterers. Likewise, more observations would also solidify the presented explanations why the magnitude and direction of the effect of artificial illumination differed between station A, B and C.

Further remarks: I suggest using a less alarmistic term for "light pollution", e.g. "artificial illumination. More minor remarks are indicated in the commented manuscript.

Hauke Flores

Reviewer #2 (Remarks to the Author):

The general comments

The manuscript considered possible effects of artificial light from research vessel on behavior of zooplankton and fishes in the Barents Sea during the Polar night. The theme of the manuscript is very interesting and is the continuation of previous investigations of the authors. Based on the results of 3 fields experiments, the authors state that artificial light change vertical distribution and behavior of zooplankton and fishes under the distance up to 200 m around the vessel during the Polar night. This effect can provide errors in stock assessment under research surveys during the Polar night. This manuscript can be interesting for wide range of biologists and principally can be published in "Communications Biology".

However methods of investigations and some conclusions seems to be rather debatable.

Methods

Despite the experiment design generally appreciate (dark phase and then light phase, but I cannot understand so big differences in its duration in all 3 stations (9, 49 and 178 minutes in dark and 13, 38 and 58 minute with light)), it would be more interesting to evaluate time of return zooplankton and fish to initial undisturbed state. It can evaluate duration of such "light pollution" and possible its effect on research survey results. If return of plankton and fish in the initial state will be fast, probably conclusions of the authors on large effect of "light pollution" can be changed. In addition, there is no indication in the text, was the vessel in drifting stage (with stopped engine) during the field experiment. If no, can the vessel noise impact on behavior of zooplankton and fish and what comparative levels of the impact of the "light pollution" and the impact of the vessel noise?

Conclusions

Conclusions of the authors on impact of the effect of "light pollution" on results of research surveys during the Polar night seems debatable.

Firstly, results of the authors (figure 2 and 3) showed that vertical distribution of plankton and fish were changed after light turn on. But in any case, even their vertical distribution (and the center of mass) changed, plankton and fish are evaluated by the echosounder. As for demersal fish, distributed in near bottom layers, their vertical distribution will not changed and their estimation by trawls will not changed too.

Secondly, during acoustic surveys - the authors state that the light affects on plankton and fish on distance up to 200 m from a vessel. Considering the speed of research vessel in research acoustic survey in the Barents Sea is approximate 10 knots, the vessel goes 200 m during only 20 seconds. Is such short time period crucial for acoustic estimation of pelagic fishes?

Thirdly, if estimation of pelagic and demersal fishes conducted by mid-water or demersal trawls, there are general rule – the length of cables for these trawls is approximately in 2 time large than depth in surveyed area (100 m at depth 50 m, 200 m at depth 100 m and 400 m at depth 200 m). So, trawl will work at least 100-200 m behind the vessel. In this case, the time of return to initial state is very important.

The title of the manuscript seems a bit incorrect. Firstly, it is not possible to understand that authors investigated effect of light from ships during the Polar night, not in other seasons. Secondly, the term "light pollution" seems to provide the negative sense of the title of this manuscript. I am not sure if it is correct – to provide estimate of any phenomenon before its consideration in a scientific article.

In addition, after continuation and development of the authors logic, if all other parts of the World Ocean are under diel light and dark changes - can the Polar Day be considered as "light pollution" too? Is the effect of the Polar day conditions on fish and zooplankton behavior similar to the Polar night effect?

What duration and power of possible "light pollution" from vessels on plankton and fish? Do the authors overestimate the effect of "light pollution" from vessels on zooplankton and fishes during Polar night?

It would be interesting to provide the general information on the depth of light penetration in marine water depend on water turbidity.

The specific comments

Abstract

Line 24 – the first sentence – it is too obvious statement.

Line 25-26 – "Yet, darkness is also the preferred 'habitat' of most marine organisms" – I think it is too strong statement. Most of marine organisms not prefer some habitats, but just live in appreciate habitat or die in inappreciate habitat.

Line 26 - Here we explore the potential effects of artificial light on organisms – on behavior only, not on organisms

Line 27 – is "the Arctic Polar Night" really the habitat?

Introduction

Line 34 – the first sentence – is it correct for both terrestrial and marine parts on the Earth?

Line 37 – "There are now few pristine dark habitats above ground" – it is too strong statement!

How it agree with the data from the previous sentence (23 % and 22 %). See e.g. the map from NASA (https://www.nasa.gov/images/content/712130main_8246931247_e60f3c09fb_o.jpg)

Line 39 – is it related to investigations on terrestrial or marine environment?

Line 51-52 – how is it related to the Polar Night conditions?

Line 76 – "and development of marine resources in the Arctic" – why?

Results and Discussion

Line 92-95 – better use station A, station B and station C, not the first, second and third stations.

Figure 1 – lines for salinity and turbidity are very similar and difficult to recognize

Lines 110-111 – "These responses were tested by turning on the ship's lights after a period of total darkness." Can the duration of the dark and light periods affected on results?

Lines 214-215 – "The potential effects of light pollution on nighttime estimates of backscatter could be significant for the assessment of many stocks of commercially exploited species." – I think changes in vertical distribution of pelagic fishes will not impact on its stock assessment by echosounder for acoustic survey. The potential effect on demersal fish looks very doubtful.

Figure 4 – does it really needed? In addition, I am not sure that all vessels and always use such powerful spotlights.

Line 237 – what id DVM, no any explanation in the text. Diel vertical migrations?

Impact on stock assessments

Most examples (surveys of pelagic fishes) referred by the authors are research surveys conducted not during the Polar night, but in other seasons with distinct day and night change. So, the problems with differences in night and days catches due to diel vertical migrations are well-known. The Polar night is the very special case, and for me possible impact of "light pollution" on research survey is overestimated by the authors.

Material and methods

Species composition

Lines 265-266 – long rough dab, not American plaice

Line 269 – what is “common shrimp”?

Fish and zooplankton sampling

Lines 316-321 – duration of trawling by midwater trawl? When were trawlings conducted – before or after experiment with lights off and on?

Line 324 – “Hydrobios”, not “Hydrobioas”

The same - when were plankton samplings conducted – before or after experiment with lights off and on?

Is it necessary to add the latin names for all fish species mentioned in the manuscript?

References

Lines 418-420 –reference 25 – is there the title of this publication

My general conclusion is that the manuscript can be accepted for publication in “Communications Biology”. If my comments will be useful for authors and the editors will agree with my comments, the comments should be taken into account for the manuscript revision.

Each comment is answered directly in the text below (red, italics), but a few issues raised are commented on separately in the beginning:

Reviewer 1, comment on sample size: We conducted three field experiments, all of them showed a clear response in relation to artificial light (both reviewers commented on the term “light pollution”, hence we have in the revised version changed this to “artificial light” as suggested by reviewer 1). The clear response by the pelagic community to light in all three experiments is the new critical observation and we acknowledge that this point did not come through well enough to the readers. By a visual inspecting of the echograms, by far strongest response was seen at station A. This extreme response, seen across the entire water column down to the very bottom at 200m was of course unexpected, and is therefore a cornerstone of the manuscript. However, quantitative analyses showed that the acoustic backscatter (NASC) in the water column did in fact change in all three experiments (between 19% and 49% at 18 kHz). It is important to stress here that the response to artificial light was different between stations, but that the experimental design was similar (the same artificial light used, boat drifting without power on the propellers, same acoustic instruments pinging on the ship and same noise created by the ship). The difference in response can hence only be attributed to either differences in species composition and/or differences in the inherent optical properties of the water that result in a changed underwater light field. Hence, and this is a point we have now tried to stress more clearly in the manuscript, our results clearly show that a) artificial light affects the marine community in the polar night and b) the quantitative and qualitative nature of the response can vary. This latter point might actually be seen as the most important part of the manuscript, as this so clearly demonstrates that the influence of artificial light is strong, but that the precise nature of the response can only be assessed and predicted based on a thorough understanding of the physical and biological characteristics of the impact area. We have tried to highlight these aspects more strongly in the revised manuscript (Abstract and start/end of the Results & Discussion section).

Reviewer 1, comment on vertical dimension: The reviewer points out the value and timeliness of the manuscript, linking it to the large ongoing MOSAiC expedition. At the same time, the reviewer also asks for a broader geographical relevance of the study, pointing out that our experiments are limited to coastal regions and the Barents Sea. We strongly agree with the reviewer that it would be relevant to also look at potential effects in the Arctic Ocean and “how pelagic deep-ocean communities would react to artificial illumination”. However, this is exactly why our manuscript is both timely and important now – the MOSAiC expedition is ongoing, the Norwegian Polar Institute carried out a similar expedition 2 years ago and the Russians plan a similar drift-expedition into the Arctic next year. Our results will be highly valuable for all of the above-mentioned expeditions when it comes to analyzing and interpreting their data, especially those collected in the upper 200m of the water column, and these will themselves provide a larger geographical relevance to the documented effects of artificial light on the distribution of pelagic biota. We also want to point to the fact that the three stations selected were carefully chosen in order to represent as large variability as possible in relation to physical conditions, in order to be relevant on a larger pan-arctic scale. We therefore argue that our results are both strong, relevant and important to publish, and that they will be actively used / cited by many papers produced from the above-mentioned expeditions, and elsewhere.

Reviewers' comments:

Reviewer #1 (Remarks to the Author):

This study addresses a highly relevant issue of marine ecosystem research and provides new data from the in this respect under-studied Arctic Ocean. I consider it a highly valuable and timely contribution to science with multiple implications, not the least on the on-going year-round ice drift study MOSAiC. The hydroacoustic methodology and the presented analyses are solid. However, the sampling of zooplankton communities is of limited value to support the hydroacoustic data, because two very different types of zooplankton nets were used. Furthermore, the fish community could not be sampled at station C.

Answer: We agree with the reviewer that the zooplankton samples are of limited value. They were merely included in the manuscript to provide insights into which species are present in relation to interpreting the acoustic data – they indicate numerically dominant taxa, not a full community characterization. It falls outside the scope of this manuscript to provide a more detailed description of the zooplankton community at the three stations. Zooplankton species composition is for this reason not discussed or presented in great detail, but strictly used to explain why the response to artificial light was different. It is unfortunate that it was not possible to trawl at station C, but at least the macrozooplankton species composition is well covered by sampling with the MIK net. Zooplankton abundance is only mentioned in the Materials section to highlight that this is not a key motivation of the manuscript – net and trawl data only serve the purpose of allowing us to discuss in general terms which species are most abundant in the water column during the three experiments, and hence were most likely contributors to the observed responses to artificial light.

My only concern with this study is that – given the differences in direction and magnitude of the postulated effect of light on pelagic animal assemblages – the sample size is relatively low, and limited to the shallow habitat of the Barents Sea / Svalbard shelf. It would be quite relevant to know the vertical dimension on this effect, and how pelagic deep-ocean communities would react to artificial illumination. Furthermore, a really strong effect was only observed at one station (Station A). To support the postulated general bias of sampling from illuminated ships, one would like to see this effect at more than one station to exclude that an unknown and unintended factor rather than light caused the observed dispersion of backscatterers. Likewise, more observations would also solidify the presented explanations why the magnitude and direction of the effect of artificial illumination differed between station A, B and C.

Answer: See two comments above. It should be pointed out that it is a misunderstanding, no doubt due to unfortunate flaws in our original text, that the general bias of sampling from illuminated ships is based on one single field experiment – it is based on the effects observed in all three experiments performed.

Further remarks: I suggest using a less alarmistic term for “light pollution”, e.g. “artificial illumination. More minor remarks are indicated in the commented manuscript.

Answer: See comment above. We acknowledge that both reviewers recommend that we do not use the term “light pollution” and have changed this to “artificial light” throughout the text.

Reviewer #2 (Remarks to the Author):

The general comments

The manuscript considered possible effects of artificial light from research vessel on behavior of zooplankton and fishes in the Barents Sea during the Polar night. The theme of the manuscript is very interesting and is the continuation of previous investigations of the authors. Based on the results of 3 fields experiments, the authors state that artificial light change vertical distribution and behavior of zooplankton and fishes under the distance up to 200 m around the vessel during the Polar night. This effect can provide errors in stock assessment under research surveys during the Polar night. This manuscript can be interesting for wide range of biologists and principally can be published in “Communications Biology”.

However methods of investigations and some conclusions seems to be rather debatable.

Methods

Despite the experiment design generally appreciate (dark phase and then light phase, but I cannot understand so big differences in its duration in all 3 stations (9, 49 and 178 minutes in dark and 13, 38 and 58 minute with light)), it would be more interesting to evaluate time of return zooplankton and fish to initial undisturbed state. It can evaluate duration of such “light pollution” and possible its effect on research survey results. If return of plankton and fish in the initial state will be fast, probably conclusions of the authors on large effect of “light pollution” can be changed.

Answer: We agree with the reviewer that a “return to...initial undisturbed state” would be highly valuable to observe, and was also a characteristic of the system that we originally aimed at quantifying. However, due to the highly patchy nature of the communities, a return to “initial undisturbed state” becomes a rather subjective concept. Instead, we included Figure 3 in order to quantify how far away from the ship we need to go before we no longer see an impact from the artificial light produced by the ship. As the field experiments had to be carried out from a drifting vessel, and at least in station A and C in areas where advection can be expected to be strong, the patchiness of the marine communities is simply too great to expect that a return to an “initial undisturbed state” is to be expected or even possible.

It should also be noted that the reason why the duration in time at the three stations were so different is simply based on the complexity of the response detected. No one could have predicted the extreme response observed at the first station a priori – with the new knowledge in hand we extended the periods of the following experiments. However, as the response was so different in station B and C compared to A, the extended duration in hindsight may seem random. We could have excluded the additional time at the latter two stations (experiments), but we feel it is better and more correct to present the data and results as they are.

In addition, there is no indication in the text, was the vessel in drifting stage (with stopped engine) during the field experiment. If no, can the vessel noise impact on behavior of zooplankton and fish and what comparative levels of the impact of the “light pollution” and the impact of the vessel noise?

Answer: We acknowledge that this is a factor that we did not explain well enough in the text, and have now updated the manuscript accordingly (first part of the Materials section). The ship was drifting with the engine on, but without any power to the propellers. The design was similar at all three stations, whereas the observed response to artificial light were different, thereby making it highly unlikely that the observed effects could be due to vessel noise. We now state this clearly in the Materials section.

Conclusions

Conclusions of the authors on impact of the effect of “light pollution” on results of research surveys during the Polar night seems debatable.

Firstly, results of the authors (figure 2 and 3) showed that vertical distribution of plankton and fish were changed after light turn on. But in any case, even their vertical distribution (and the center of mass) changed, plankton and fish are evaluated by the echosounder. As for demersal fish, distributed in near bottom layers, their vertical distribution will not change and their estimation by trawls will not change too.

Answer: The reviewer is correct in that the vertical distribution of plankton and fish were estimated using acoustics, and that the conclusions need to be limited to what acoustics are able to tell us. However, the acoustic methods used are highly conventional and commonly used across many fields of marine science. In fact, the acoustic methods used are the go-to method for biomass estimates of pelagic fish. We therefore do not really see how this comment could be taken into effect regarding the pelagic community. However, we acknowledge that we have not been clear enough in our original text, as we never intended to say that demersal fish change their vertical distribution. This is a misunderstanding that we have tried to clarify in the text. However, we are uncertain where this misunderstanding actually stems from -

Secondly, during acoustic surveys - the authors state that the light effects on plankton and fish on distance up to 200 m from a vessel. Considering the speed of research vessel in research acoustic survey in the Barents Sea is approximate 10 knots, the vessel goes 200 m during only 20 seconds. Is such short time period crucial for acoustic estimation of pelagic fishes?

Answer: As shown on figure 2, the response from the organisms to artificial light from a vessel is instantaneous (<5 sec), most likely due to a rapid change in orientation resulting in lower (or higher) backscatter. Hence, 20 seconds is more than enough to result in a significant variation in backscatter

on the hull-mounted echosounder and to bias acoustic estimates. We now state how quick the response is on page 4. This is now included in the beginning of the “Impact on stock assessment” paragraph.

Thirdly, if estimation of pelagic and demersal fishes conducted by mid-water or demersal trawls, there are general rule – the length of cables for these trawls is approximately in 2 time large than depth in surveyed area (100 m at depth 50 m, 200 m at depth 100 m and 400 m at depth 200 m). So, trawl will work at least 100-200 m behind the vessel. In this case, the time of return to initial state is very important.

Answer: As we demonstrate in figure 3, the light footprint of the vessel (i.e. the horizontal distance from the vessel at which light has an impact) can reach 200 m, which is within the distance the trawl would be from the ship. Yet, we demonstrate that it is mainly the backscatter that changes rapidly, not the centre of mass. This suggests a change in orientation rather than a rapid escape and thus limited impact on trawl catches but strong impact on acoustic estimates. Hence, we suggest that the impact of artificial light on fish stock assessment surveys mainly apply to pelagic species, which are based on acoustic surveys more than trawl (due to trawl avoidance). The impact on demersal fish stock assessment is likely not as important. This is now clarified in the last part of the discussion by emphasising the word “pelagic”.

The title of the manuscript seems a bit incorrect. Firstly, it is not possible to understand that authors investigated effect of light from ships during the Polar night, not in other seasons. Secondly, the term “light pollution” seems to provide the negative sense of the title of this manuscript. I am not sure if it is correct – to provide estimate of any phenomenon before its consideration in a scientific article.

Answer: We have updated the title in accordance with the reviewer’s suggestion

In addition, after continuation and development of the authors logic, if all other parts of the World Ocean are under diel light and dark changes - can the Polar Day be considered as “light pollution” too? Is the effect of the Polar day conditions on fish and zooplankton behavior similar to the Polar night effect?

Answer: We do no longer use the term “light pollution”, and hence consider this comment now outdated.

What duration and power of possible “light pollution” from vessels on plankton and fish? Do the authors overestimate the effect of “light pollution” from vessels on zooplankton and fishes during Polar night?

Answer: Unless we misunderstand the comment from the reviewer, the data asked for by the reviewer is in fact presented in both Table 2 and figure 3. We also state in the result section what the quantitative “power” of the artificial light is, as well as providing a quantification of light at depth throughout the entire water column. This quantification is based on direct measurements of IOPs.

It would be interesting to provide the general information on the depth of light penetration in marine water depend on water turbidity.

Answer: Provided in Table 2 and Figure 1 (turbidity)

The specific comments

Abstract

Line 24 – the first sentence – it is too obvious statement.

Line 25-26 – “Yet, darkness is also the preferred ‘habitat’ of most marine organisms” – I think it is too strong statement. Most of marine organisms not prefer some habitats, but just live in appreciate habitat or die in inappreciate habitat. **The first lines in the abstract are updated according to the comment**

Line 26 - Here we explore the potential effects of artificial light on organisms – on behavior only, not on organisms. **Text updated accordingly**

Line 27 – is “the Arctic Polar Night” really the habitat? **Changed “habitat” to “environment”**

Introduction

Line 34 – the first sentence – is it correct for both terrestrial and marine parts on the Earth? **Globally yes, but logically limited to coastal regions (not offshore)**

Line 37 – “There are now few pristine dark habitats above ground” – it is too strong statement! How it agree with the data from the previous sentence (23 % and 22 %). See e.g. the map from NASA (https://www.nasa.gov/images/content/712130main_8246931247_e60f3c09fb_o.jpg) This sentence is now deleted

Line 39 – is it related to investigations on terrestrial or marine environment? yes

Line 51-52 – how is it related to the Polar Night conditions? It is not entirely clear what the reviewer actually mean here, hence we have not changed the text. But if the question posed is related to whether artificial light is relatively new phenomenon in the Arctic marine polar night, then the answer is “yes it is”.

Line 76 – “and development of marine resources in the Arctic” – why? Text modified to make our point more clear (development of marine resources was too vague)

Results and Discussion

Line 92-95 – better use station A, station B and station C, not the first, second and third stations.

Figure 1 – lines for salinity and turbidity are very similar and difficult to recognize Changed accordingly

Lines 110-111 – “These responses were tested by turning on the ship’s lights after a period of total darkness.” Can the duration of the dark and light periods affected on results? Theoretically yes, but as the main point of investigation was to examine if artificial light actually does affect the behavior of fish and zooplankton, such a temporal aspect falls outside the scope of this study.

Lines 214-215 – “The potential effects of light pollution on nighttime estimates of backscatter could be significant for the assessment of many stocks of commercially exploited species.” – I think changes in vertical distribution of pelagic fishes will not impact on its stock assessment by echosounder for acoustic survey. The potential effect on demersal fish looks very doubtful. Here we actually disagree with the reviewer, but we acknowledge that our original text did not made our point clear enough; as we argue that the immediate response is most likely due to a change in orientation, it is not the vertical distribution as such that changes, but rather the immediate change in backscatter as seen on the echograms. As we point out in figure 2, the *backscatter* (a proxy for animal density and biomass) changes immediately after the light is turned on. The actual vertical distribution most likely change much less, but the interpretation of the echograms, had this been in a stock assessment survey, would have resulted in a much lower or higher biomass estimate.

Figure 4 – does it really needed? In addition, I am not sure that all vessels and always use such powerful spotlights. No, this image is not necessary, and if the editor insists we are willing to remove it. It does show what a normal working light situation looks like in the high Arctic during the polar night – the strong lights will most cases be on in order to look for sea ice. But the reviewer is correct that is not totally critical to the manuscript

Line 237 – what id DVM, no any explanation in the text. Diel vertical migrations? Yes, diel vertical migration. This is now spelled out on the text.

Impact on stock assessments

Most examples (surveys of pelagic fishes) referred by the authors are research surveys conducted not during the Polar night, but in other seasons with distinct day and night change. So, the problems with differences in night and days catches due to diel vertical migrations are well-known. The Polar night is the very special case, and for me possible impact of “light pollution” on research survey is overestimated by the authors.

The reviewer is correct in that the polar night is a special case and that most cases referred to are not carried out in the polar night. We also agree with the reviewer that problems with DVM are well known and taken into account. However, and this is a key point, organisms performing DVM are by default light sensitive. Hence, and even if you take into account the strong diurnal changes in vertical position, surveys carried out by night might still potentially impose a bias in the stock assessments, and cannot be ruled out without testing the effect of artificial light. This last point is what we try to stress. Moreover, what we suggest here is that acoustic estimates, not catches, will be the most biased.

Material and methods

Species composition

Lines 265-266 – long rough dab, not American plaice. **corrected**

Line 269 – what is “common shrimp”? **changed to “northern shrimp”**

Fish and zooplankton sampling

Lines 316-321 – duration of trawling by midwater trawl? When were trawlings conducted – before or after experiment with lights off and on? **Before, now included in the text**

Line 324 – “Hydrobios”, not “Hydrobioas” **corrected**

The same - when were plankton samplings conducted – before or after experiment with lights off and on? **In the dark during the experiments. Unlike the trawling, which for safety reasons could not be carried out in the dark, all net samples were carried out with all lights turned off. This is now included in the text**

Is it necessary to add the latin names for all fish species mentioned in the manuscript? **We prefer to include the latin names, as some of the common names actually differ between North America and Europe for the same species (Boreogadus saida is one good example – polar cod in Europe and arctic cod in North America).**

References

Lines 418-420 –reference 25 – is there the title of this publication. **Book title is included**

REVIEWERS' COMMENTS:

Reviewer #1 (Remarks to the Author):

The revised manuscript constitutes a significant improvement compared to the initial version. In particular, the authors toned down some of their language and made it clear that their results show that pelagic animals do react to light, but that the nature of this reaction is not unidirectional. The authors answer my main concern regarding the low sample size and differing responses of the 3 communities to light by arguing that the differences in the 3 sites represent a large variety of environmental conditions and therefore the results can be generalised, even to deep-sea situations. I would still be very cautious with this interpretation, as such highly variable results in a geographically confined shallow sea may imply even more variable behaviour in the deep ocean. So I am afraid that even in this improved version my concern about the low sample size is still valid to some extent. With these data, we cannot be sure whether the observed variability in behavioral response was due to the environment and the taxonomic composition, as suggested by the authors, or simply a statistical variation. Therefore, I would support publication, but I must leave it to the editor to decide whether the results are solid enough for *Communications Biology*, or better suited in a more specific journal. Regardless of the journal it will be published in, I am now even more convinced that these data should be published. I agree with the authors that the study will be highly cited and is relevant to various past and on-going polar night studies.

Reviewer #2 (Remarks to the Author):

Generally, all issues related to my questions on methodology and some small comments on the text were revised by the authors, so there are no problems.

However some general questions remain the same. I see that some questions (e.g. investigations of time needed for returning in undisturbed state) cannot be solved in the manuscript without field works.

But the conclusion on impact of artificial light on results of surveys and possibly stock assessments seems to be too strong - surveys of all mentioned fish species (capelin, herring, blue whiting) are not conducted in the area where the Polar night is present or not during the Polar night. Impact of artificial light on results of survey of demersal fishes is probably absent - as the authors noted in their reply, but not mentioned in the manuscript.

Generally, the manuscript is interesting. So, if editors consider that my concerns too hypercritical, the manuscript can be accepted for publication.

Reviewer 1.

Sample size and relevance for other regions: We are pleased to see that the reviewer acknowledge that although we were only able to conduct three field experiments, the three sites represent very different biological and physical habitats. We agree that we should be careful not to extrapolate the results found at these three sites to a too broad geographical area. Our text in this aspect is therefore strongly toned down – we now raise this as an open question in the fourth paragraph of the Discussion. The final concluding paragraph in the Discussion is now limited and focused only on the Arctic polar night (not a broader geographical region).

Reviewer 2.

General questions regarding recovery time: We very much agree with the reviewer, but the nature of the experiments and the physical conditions (advection of biological communities, drift of the vessel, natural patchiness) in the investigated regions did not allow for investigating recovery time. This remain an important question that further research need to look into.

Impact of artificial light on stock assessment: We acknowledge that our manuscript may have been too speculative on this matter, and have now toned this aspect considerable down. It is still mentioned, but more as a question raised than a conclusion.

Impact restricted to pelagic, not demersal organisms: In the results section, the second subheading now clearly specifies that we are looking at effects on pelagic organisms, not demersal. It was never our intention to say that we observe any effect on demersal organisms, and we hope this should now have been clear for any reader.